# Exploring Institutional Pressures, Green Innovation, and Sustainable Performance: Examining the Mediated Moderation Role of Entrepreneurial Orientation

**Qiang Zhang [1], Xiumei Zhu [1] and Min-Jae Lee [2],***

[1] College of Economics and Management, Weifang University of Science and Technology, Weifang 262700, China; zhangqiang@wfust.edu.cn (Q.Z.); zhuxiumei@wfust.edu.cn (X.Z.)
[2] Department of Global Business, Mokwon University, Daejeon 35349, Republic of Korea
* Correspondence: mjlee@mokwon.ac.kr

**Abstract:** This study aims to understand the multifaceted role of entrepreneurial orientation between institutional pressures, green innovation, and sustainable performance by using institutional theory and the entrepreneurship perspective as a comprehensive theoretical lens. To be more specific, this study not only analyzes the impact of institutional pressures consisting of regulatory, normative, and cognitive pressures on green innovation and the mediating effect of entrepreneurial orientation but also examines the moderating effect of entrepreneurship between green innovation and sustainable performance. Empirical results based on survey data from 483 listed firms in China indicate a positive effect between institutional pressures and green innovation and confirm the mediating effect of entrepreneurial orientation. Meanwhile, between green innovation and sustainable performance, entrepreneurial orientation showed a significant negative moderating effect. Our findings show that institutional pressures can drive corporate green innovation and suggest that entrepreneurial orientation can help achieve green innovation by encouraging them to challenge more innovative environmental practices based on institutional pressure. On the other hand, in firms that have not had enough green innovation, a high entrepreneurial orientation can undermine sustainable performance because it can increase risk.

**Keywords:** institutional pressures; green innovation; entrepreneurial orientation; sustainable performance; mediated moderation role

## 1. Introduction

As the severity of environmental pollution increases, various stakeholders, including consumers and governments, are gradually interested in firms' efforts to achieve sustainable growth that can eventually contribute to economic and social development while protecting the natural environment [1,2]. Harmonizing environmental, social, and economic performance is key to sustainable growth, and these three dimensions represent a "triple bottom line" that looks at corporate performance [3]. In response to these growing external pressures from stakeholders, many firms aim to reduce damage to the environment while improving corporate sustainable performance based on green innovation [4]. Yang et al. [5] document that the adoption and implementation of green innovation can help businesses reduce their environmental burden and contribute to economic development and building an efficient social system. Our study aims to identify the mechanisms by which firms drive green innovation to achieve sustainable performance.

Green innovation refers to processes that contribute to the development of production and operation with the aim of reducing environmental risks, such as the negative consequences of the natural environment [6]. Green innovation can make major changes in the way firms do business by effectively controlling pollution and resources, but it requires

more financial input than other environmental practices (i.e., behaviors for environmental preservation) [7]. Nevertheless, some scholars argue that since green innovation has essential therapeutic effects on improving sustainable issues [8], it is necessary to design institutional devices to impose adequate pressures on firms and further urge them to actively carry out environmental practices [9].

While there is a wide consensus about the need to make efforts to improve environmental practices, at least from an institutional perspective [10,11], little is known about why some firms engage in more green innovation than others. This theory explains that firms will use similar environmental practices due to the pressures formed by the institution and will be "isomorphic" with each other [12,13]. However, as firms vary in the consensus of environmental practices that are actually implemented under similar institutional pressures, there remains a gap between theory and practice. It also focuses on explaining the link between green innovation and corporate performance [14,15], but the empirical results are conflicting.

There can be three reasons for this inconsistency. First, this is because institutional pressures on green innovation by firms are complex concepts composed of multiple dimensions [4]. These institutional pressures are divided by Scott [16] into three dimensions: regulatory, normative, and mimetic pressures, and it should be taken into account that even similar institutional environments can lead to institutional recognition differences. As institutional recognition can define the firm's purpose and existence [10], institutional pressures can strongly shape the adoption of green innovation within an organization [11]. In particular, while existing studies from an institutional perspective have highlighted the effects of regulatory and normative pressures in the environmental context [4,12], mimetic pressure has largely been ignored. However, as environmental practices play a pivotal role in the survival and sustainability of a firm, there is increasing pressure to emulate the examples of leading firms that achieve successful innovation results based on their environmental strategies [17]. Thus, drawing on insights from institutional theory and environmental innovation literature, we argue that specific dimensions, such as regulatory pressure, normative pressure, and mimetic pressure, make green innovation more facilitated by focal firms.

Second, this reason may be that the focal relationship of green innovation to institutional pressures depends heavily on entrepreneurial orientation. Entrepreneurial orientation is an organizational phenomenon in which top management seeks to create new value through challenges and innovation activities [18]. Not all firms facing similar institutional pressures seek green innovation in response to these pressures. Innovation inherently involves uncertainty, and businesses are struggling to implement green innovation because they need to control limited resources while increasing their success rates [19]. In particular, the characteristics of some top management greatly influence a firm's decision-making with regard to participation and investment in green innovation practices [20]. The entrepreneurial orientation of top management to adopt corporate strategies and reconfigure resources can be a pivotal driver of green innovation [21]. Thus, there is increasing evidence that entrepreneurial orientation is important not only for the economic prosperity of nations but also for corporate survival and growth [22]. However, current research does not consider how institutional contexts provide entrepreneurial orientation with opportunities for green innovation. Thus, exploring the relationship between institutional pressures, entrepreneurial orientation, and green innovation assumes significant importance, and this study aims to bridge this gap by exploring how entrepreneurial orientation mediates environmental innovation based on institutional pressures.

Third, since extensive investment within the organization is required to sustain environmental innovation, it is necessary to understand the relationship between entrepreneurial orientation and corporate performance in more depth. While stakeholders are increasingly emphasizing environmental practices, corporate top executives are seen as hesitant to embrace green innovation as they weigh the benefits and losses of their environmental practices [23]. Existing views on the impact of entrepreneurial orientation

on corporate performance vary across both positive [24] and negative perspectives [25]. This can predict that entrepreneur orientation may moderate the impact of green innovation adoption on corporate performance. In addition, while one of the main goals of green innovation is to change the constitution of an organization to minimize its impact on the environment and achieve sustainable performance through the development of better products and services, to the best of our knowledge, no studies have examined the role of entrepreneur orientation in a sustainable management context. Thus, this study aims to contribute to the expansion of knowledge by considering entrepreneurial orientation as a potential moderator in the relationship between green innovation and sustainable performance.

We address this interest through three research questions: (1) What institutional pressures support green innovation? (2) Does entrepreneurial orientation mediate the pursuit of green innovation under institutional pressures? (3) What is the moderating effect of entrepreneurial orientation on the relationship between green innovation and sustainable performance?

This study makes several contributions to fill the research gaps in the current literature. First, this study contributes to expanding the discussion of Porter and Van der Linde [26] and enriching institutional theory by examining the various dimensions of institutional pressures and their relationship to green innovation. Second, this study examines the mediation effects of entrepreneurial orientation that drive firms toward green innovation under institutional pressures, thereby linking institutional perspective with entrepreneurship theory to strengthen our knowledge of the core drivers of green innovation. Third, it further enriches the sustainable management literature by suggesting that entrepreneurial orientation plays a moderating role in the relationship between green innovation and sustainable performance in a business environment, where environmental issues are becoming increasingly serious.

## 2. Theoretical Background

### 2.1. Institutional Perspective on Environmental Sustainability

The institutional perspective has been useful in examining the adoption and diffusion of corporate organizational practices formed on the basis of norms, values, and cultures established in a specific society [27,28]. Institutional theory explains that organizations accept institutional environments in pursuit of legitimacy, which puts significant pressure on organizational behavior [29]. Meyer and Rowan [30] suggest that organizational practices are deeply ingrained in and reflect a widespread understanding of social reality enforced by public opinion, by the views of important constituents, by knowledge legitimated through the educational systems, by social prestige, and by the laws. Thus, proponents of this theory explain that firms will use similar practices due to the pressures formed by the institution and will be "isomorphic" with each other [12,13]. Previous studies have found that as institutional pressures increase, firms become increasingly similar in their quest for legitimacy [31]. Firms attempt various actions to strengthen or protect the legitimacy of their business under institutional pressures [4]. Meanwhile, due to the recent seriousness of environmental pollution, the social atmosphere positively evaluates the introduction of various systems to reduce environmental impact [32]. The increasing stringency of environmental issues makes green innovation more attractive, with firms adopting environmental practices to secure legitimacy [33]. For example, the Chinese government is actively implementing environmental policies and strengthening enforcement, and firms are increasingly participating in green innovation initiatives [34].

### 2.2. Green Innovation

Green innovation aims to reduce pollution through the development of products, services, processes, and methods that can promote resource conservation, implement clean energy alternatives, and reduce waste emissions [35]. As environmental issues emerge as a topic of corporate survival, many firms are promoting green innovation as a tool to

reduce the negative impact on the environment and positively impact corporate competitive advantage [36]. Chen et al. [37] explain that green innovation generally involves green product/service innovation and green process innovation. Specifically, green product/service innovation means reducing the negative effects of environmental pollution and waste of resources by considering environmental factors when providing new products and services. Green process innovation refers to efforts to reduce resource waste and achieve environmental improvement by integrating environmental issues and green technologies into existing processes. Many scholars emphasize that green innovation can reduce the negative impact of activities on the environment and resources and increase sustainability by developing and utilizing new products, services, and processes [38,39].

Previous research reports that firms seeking environmental innovation reduce pollution rates and increase recycling [40], mitigating environmental problems [41] while saving energy and improving resource efficiency [42]. In addition, green innovation helps firms differentiate their business models from their competitors [36] and is also essential to maintaining legitimacy [43]. However, as firms often need huge financial investments to drive green innovation, it is not enough to lead green innovation voluntarily; thus, incentives and pressures from governments and other institutions are required [44]. Therefore, institutional theory provides an appropriate theoretical lens for considering corporate green innovation [45].

### 2.3. Entrepreneurial Orientation

Starting with Schumpeter's mention that entrepreneurship is an important driving force for innovation, many studies have examined the relationship between entrepreneurship and corporate innovation [46]. In this vein, entrepreneurial orientation has evolved into one of the most studied topics in the entrepreneurship literature and has become an important strategic orientation for corporate growth and survival by applying corporate-level phenomena [47]. However, entrepreneurial orientation is different from entrepreneurship itself. Although entrepreneurship simply refers to new entry, a firm's entrepreneurial orientation refers to the entrepreneurial process, namely how entrepreneurship is undertaken [22]. Entrepreneurial orientation is reflected in the top management's perception of a firm's overall strategic decisions, which are expressed in policy, either formally or informally [48]. In particular, drivers of green innovation include support for top executives and environmental commitments [49]. Prior research indicates that the adoption of green innovation by top management can be strengthened by institutional factors such as government regulation [50], normativity [51], and stakeholder engagement [52]. Thus, entrepreneurial orientation includes a corporate strategic intention to continue and deliberately leverage opportunities for growth [53] and can have a significant impact on decision-making and resources related to green innovation [54].

### 2.4. Sustainable Performance

Sustainable performance means the harmonious growth of the three pillars of sustainable development (i.e., economic, social, and environmental), called the triple bottom line. Elkington [3] suggests that three sustainability dimensions should produce balanced performance to achieve sustainable development. Due to the environmental crisis, sustainable performance is becoming increasingly important for long-term survival and competitive advantage for firms [55,56]. In this vein, Kamble et al. [57] document that green practices such as green purchases, green manufacturing, green information systems, and ecological design have a positive impact on corporate sustainability. In addition, Khan et al. [58] point out that green innovation mitigates the negative impact on environmental sustainability in a firm's production operations.

## 3. Hypothesis Development

### 3.1. Institutional Pressures and Corporate Green Innovation

The external institutional context in which a firm is embedded limits the corporate operation sphere and influences its strategic response [59]. Firms try to match strategies and behaviors with the expectations of institutions through isomorphism [11]. Isomorphism plays an essential role in the way an institution functions and is the 'pressure' that the core mechanism by which institutional isomorphism occurs puts on firms [60]. According to the premise of institutional theory [27,61], there are differences in pressures, and the relevance may vary depending on the situation. In particular, a number of studies examining environmental practices emphasize the need for special attention to consider two national dimensions that clearly define how different countries regard and respond to environmental issues: the regulatory and normative dimensions [12,13]. In addition, as business environment uncertainty and environmental issues grow, more and more firms recognize and mimic the actions of competitors to obtain legitimacy in response [17]. As such, the correlation between the three types of institutional pressure and green innovation varies in context, and this study examines the impact of these dimensions of institutional pressure on green innovation.

First, regulatory pressure includes the existing laws and rules, compensation, and even sanctions in certain national environments that promote certain types of behavior and restrict others [62]. While traditional economic scholars believe that government environmental regulations can hinder innovation by increasing corporate environmental costs [63,64], more studies, including Porter and Van der Linde [26], explain that proper and flexible environmental regulations can rather empower corporate green innovation [44,65]. Environmental regulations allow firms to use them as tools for corporate reform by emphasizing inefficiencies in resources and potential technological improvement opportunities, raising environmental awareness among members, and creating pressure to stimulate innovation and progress [66].

Some empirical studies have shown that environmental regulations benefit green innovation. Eiadat, Kelly, Roche, and Eyadat [64] stated that environmental regulations help firms overcome organizational inertia and challenge green innovation activities such as clean technology development. In addition, Menguc et al. [67] demonstrated that if governments strictly oversee pollutant emissions based on laws and regulations and combine administrative and criminal penalties, businesses can engage in environmental innovation to avoid both political and economic risks. As such, regulatory pressure can have a stimulating effect of "carrot and whip" on firms, thereby serving as a motivation to drive green innovation. Thus, we propose the following hypotheses:

Second, normative pressure stems from collective expectations, values, and standards within a particular organizational environment [27]. In the form of this pressure, social norms and stakeholder demands drive firms to respect relevant environmental regulations and engage in green innovation to meet environmental requirements [68,69]. Indeed, market demands are often considered important motivations for firms to implement green innovations [70]. As society becomes more aware of the environment, the market environment is putting more and more pressure on businesses to take environmentally responsible actions [71]. In this vein, Delmas and Toffel [72] revealed that close cooperation with market players further promotes organizational innovation on environmental issues. Berrone, Fosfuri, Gelabert, and Gomez-Mejia [4] emphasized that the concerns of stakeholders related to environmental issues have a positive impact on firms' adoption of green innovation. Li [73] argued that green consumerism is the strongest pressure to induce green innovation. Moreover, Wang, Li, and Zhao [12] found that increasingly strengthened environmental norms spur firms to perform green innovation.

Third, as sustainability emerges as the mainstream of the business environment, there is a movement to adopt green innovation practices by imitating the actions of colleagues who are creating results through green innovation. The more uncertainty or turmoil in the business environment, the more firms tend to try to reduce the risk of decision-making

by imitating the behavior of the most successful firms in the industry. In order to respond to this movement, firms may voluntarily imitate leading firms or sympathize with the environmental atmosphere. Indeed, some studies explain that the uncertainty of green innovation is the reason why green innovation is not accepted, or firms are hesitant despite its importance to a firm's competitive advantage [74,75]. As more firms adopt green innovation practices, these uncertainties and risks may be reduced as green innovation networks take shape, but barriers to entry for firms that do not accept green innovation will affect their competitiveness [76]. Eventually, peer pressure can act as a potential antecedent in determining corporate green innovation. Thus,

**Hypothesis 1.** *Institutional pressures are positively related to corporate green innovation.*

### 3.2. Mediation of Entrepreneurial Orientation

Firms with a high level of entrepreneurial orientation are known to be innovative and proactive in developing new products and services to remain competitive in a turbulent business environment [77]. However, the degree to which a company is innovative and takes risks is often affected by the nature of the institutional environment in which it operates. Wang et al. [78] argue that the impact of entrepreneurial orientation on performance depends on the level of cognitive, regulatory, and normative legitimacy of the firm. The institution can provide incentives and constraints to firms, as they can be the rules of the game [79]. Thus, entrepreneurial orientation is seen as a key factor in helping firms design environmental strategies and practices based on institutional pressures as well as offset negative environmental impacts [80].

A prior study argues that as institutions pressure them to pay attention to environmental issues, they will encourage firms to adopt entrepreneurial orientations that can help them produce innovative products and achieve sustainable development. For example, Bokusheva et al. [81] pointed out that to obtain green certification established by policies, laws, and regulations, many firms expand investments related to green innovation. Zhu and Geng [82] stated that institutional pressures facilitate the adoption of business practices to create innovation and serve as a foundation for innovation performance. Other studies have also confirmed that institutional pressures foster green innovation by stimulating corporate environmental orientation [83,84]. Based on these considerations, this study assumes that entrepreneurial orientation is a potential mediator influencing the relationship between institutional pressures and green innovation. Accordingly, the following hypothesis is proposed:

**Hypothesis 2.** *Entrepreneurial orientation plays a mediating role between institutional pressures (i.e., regulatory, normative, and mimetic pressures) and corporate green innovation.*

### 3.3. Green Innovation and Sustainable Performance

Existing environment management literature generally argues that green innovation plays a pivotal role in improving corporate overall performance by meeting the green needs of stakeholders, improving efficiency, and reducing costs [85]. Proponents explain that green innovation can help firms improve their products and internal processes, increase efficiency, and reduce operational costs to boost economic outcomes [55,56]. In addition, Grewatsch and Kleindienst [86] found that corporate eco-friendly innovation practices have a positive impact on financial performance. However, some studies advise that the potential cost increases from green innovation activities (e.g., getting green certification, green technology investment, and conversion costs of clean production) can negatively impact corporate economic outcomes [87]. In this regard, Holzner and Wagner [88] explain that while green innovation can lead to long-term and competitive gains, initial investments and costs can affect short-term profitability.

In addition, firms have recently become more conscious that green innovations related to processes and products can affect social and environmental performance [14,15]. Zailani

et al. [89] emphasize that firms that actively participate in green innovation can improve social performance by meeting stakeholder expectations. Huang et al. [90] document that top management's commitment to human capital growth and knowledge streams can be used as an essential component of green innovation that increases social sustainability. Horbach et al. [91] explain that consumer preferences are changing and that they are increasingly willing to spend more on green and green products to improve environmental performance and process innovation that is reducing energy consumption, waste, and pollution.

Moreover, green innovation can motivate organizations to create environmentally friendly products that drive environmental sustainability and increase organizational environmental performance based on efficient resource utilization [92]. Fernando et al. [93] propose that green innovation promotes the development of organizational processes and production technologies that reduce the negative impact on the environment, reduce pollution, and ultimately improve corporate sustainability. Several studies, including Chen, Lai, and Wen [37], advise that green innovation can be linked to a corporate environmental management program to minimize production waste and improve environmental performance [94,95]. Through these discussions, we argue that green innovation is positively relevant to achieving sustainable performance and propose the following hypothesis:

**Hypothesis 3.** *Green innovation is positively related to corporate sustainability performance.*

*3.4. Moderation of Entrepreneurial Orientation*

Entrepreneurial orientation plays an important role in driving green practices because it involves allocating resources for environmental practices and implementing changes in business activities [80]. Previous studies have identified entrepreneurial orientation as a major internal condition for firms that influences the relationship between innovation practices and corporate performance. Khan, Ameer, Bouncken, and Covin [53] highlight the importance of entrepreneurial orientation in determining whether firms truly embrace green practices. Wijethilake and Lama [96] argue that entrepreneurial orientation is a core indicator for successful implementation because green innovation requires significant investment and change. However, there are also studies that are skeptical of this. Tang et al. [97] point out that entrepreneurial orientation does not always have a positive impact on corporate performance but, rather, can have a negative impact when innovation is above a certain level. In the following, we argue that entrepreneurial orientation moderates the link between green innovation and corporate sustainability performance.

First, entrepreneurial orientation can achieve greater profits and performance by challenging novelty beyond thinking buried in traditional management practices (e.g., the most important purpose of a firm is to maximize profits) [98], but if excessive, the effect is likely to be negative. Tang and Tang [99] explain that a high entrepreneurial orientation may lead to confusion in decision-making, which may negatively affect corporate performance. Second, entrepreneurial orientation can create a work environment that encourages innovation and tolerates the failure and uncertainty of green innovation, and these firms are more likely to find opportunities for product improvement and pollution reduction, thereby gaining greener processes and products [100]. However, high risk sensitivity is more likely to prevent other resources from being leveraged, which can be an obstacle to the survival and growth of a firm.

As such, entrepreneurial orientation reflects top management expectations and concerns about the environmental practices and sustainable development of the firm. Thus, we postulate that entrepreneurial orientation will play a regulatory role in green innovation and sustainable performance relationships.

**Hypothesis 4.** *Entrepreneurial orientation plays a moderating role between green innovation and sustainable performance (i.e., economic, social, and environmental performance).*

Figure 1 depicts the framework of this research.

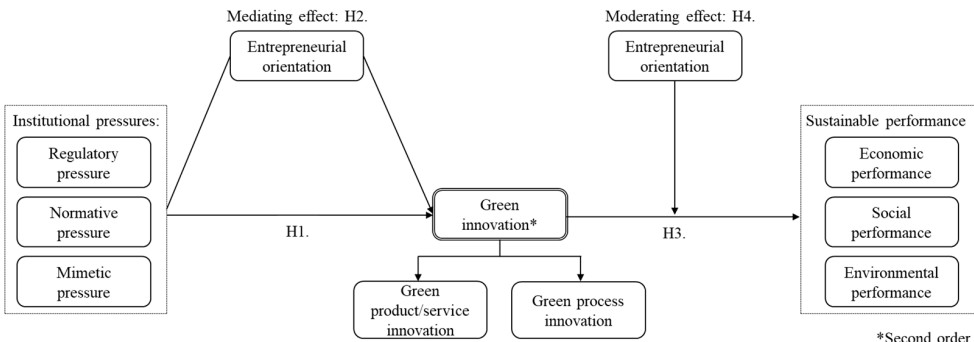

**Figure 1.** Research model.

## 4. Research Methods and Results

### 4.1. Research Context and Sample Collection

The target for hypothesis verification in this study is a listed firm in China, and the reasons for selection are as follows: China succeeds in rapid economic development based on government-centered industrial policies, but it is also criticized for making environmental pollution worse by indiscreet development [101,102]. In light of the prevailing circumstances, it is evident that the Chinese government has a strong dedication to the preservation of the environment and implements institutionally strict environmental regulations and policies [34]. In this time of industrial transformation, publicly traded firms that are highly influential in the business ecosystem face a range of challenges and changes to engage in green innovation. In this vein, the Chinese government announced "enterprise environmental information disclosure management measures" and made it mandatory for listed firms to disclose information in terms of climate change, ecological protection, and environmental protection, triggering top management's movements for green innovation. Therefore, China provides a rich context in which to examine how institutional pressures impact corporate digital orientation and sustainable performance.

By the conclusion of 2022, the China Stock Exchange had recorded a cumulative count of 5067 enterprises listed, and this study conducted a 'questionnaire survey' of these listed firms to collect data. The questionnaire was developed in English and then translated into Chinese by a bilingual researcher, and we asked a professor who had considerable research experience in the relevant field and was fluent in both languages to review it. This study requested the relevance and completeness of the items included in the questionnaire from the managers of five firms before applying the formal questionnaire procedure, and they were finalized by reflecting their correction requirements.

We asked the market research agency for a survey to collect data as quickly and accurately as possible. In a list of listed firms provided by market research institutes, the authors randomly selected 1000 potential samples, focusing on ISO 14001-certified firms: ISO-14001 [103] is an international standard for environmental management systems applicable to all industries and activities, potentially identifying corporate green innovation. The study selected top managerial staff (e.g., CEOs, vice presidents, and general managers) with extensive organizational responsibilities and access to corporate-level information as survey targets, fully explained the purpose, necessity, and confidentiality of this study, and encouraged them to participate in the survey using fixed-line telephones and e-mails. The survey spanned approximately four months, from the beginning of June to the end of September 2023, with a total of 483 significant responses (48.3% response rate).

Table 1 summarizes the profiles of the responding firms. Among the samples, manufacturing industries (N = 238, 49.3%), service industries (N = 133, 27.5%), and others (N = 112, 23.2%) responded. The size of most of the firms is 100 to less than 399 (N = 200, 41.4%), and those more than 40 years of age account for 36.0% (N = 174) of the total samples. To assess non-response bias [104], we compared firm sizes, firm age, and industry type of the responding and non-responding firms via t-tests and found no significant differences ($p < 0.05$). Thus, we confirm that non-response bias is not a serious concern for our sample.

**Table 1.** Demographic characteristics of the study sample.

| Variable | Frequency (N) | Percentage (%) |
|---|---|---|
| Industry type | | |
|    Manufacturing | 238 | 49.3 |
|    Service industry | 133 | 27.5 |
|    Others (construction, transportation, energy, etc.) | 112 | 13.2 |
| Firm size (Number of full-time employees) | | |
|    Less than 100 people | 89 | 18.4 |
|    100–399 | 200 | 41.4 |
|    400–699 | 65 | 13.5 |
|    700–999 | 85 | 17.6 |
|    More than 1000 people | 44 | 9.1 |
| Firm age | | |
|    Less than 10 years | 57 | 11.8 |
|    10–19 | 38 | 7.9 |
|    20–29 | 44 | 9.1 |
|    30–39 | 170 | 35.2 |
|    More than 40 years | 174 | 36.0 |
| N | 483 | 100.0 |

*4.2. Variables and Measurement*

All questionnaire items used in this study were adopted from the previous literature and were slightly modified to fit the context of the current study. Each item is measured on a 7-point Likert scale ranging from 'one (strongly disagreed)' to 'seven (strongly agreed)'. The explanation of the variables used in the development of the questionnaire is as follows.

First, to measure institutional pressures, measurement items were designed based on the research of Colwell and Joshi [13] and measured through four questions for each of the three dimensions (i.e., regulatory, normative, and mimetic pressures).

Second, green innovation was divided into two dimensions: green product/service innovation and green process innovation, based on the research of Chen, Lai, and Wen [37]. Specifically, we used four items to measure how firms transition to eco-friendly products and services, and four items were used for the degree of process improvement to reduce pollution and waste of resources.

Third, entrepreneurial orientation was questioned about how much companies are willing to take risks for innovation through four questions based on the characteristics of entrepreneurial orientation described by Dess and Lumpkin [105].

Fourth, sustainable performance was asked about three dimensions: economic performance, social performance, and environmental performance compared to competitors in the same industry by referring to a questionnaire from Dey et al. [106].

Finally, we controlled the impact of firm-level variables on sustainable performance. Previous studies point out that firm size, firm age, and industry type have a crucial influence on corporate performance [4,107]. Thus, we used firm size, firm age, and industry type as control variables.

Detailed components and items are presented in Table 2.

**Table 2.** Measurement and factor loadings.

| Constructs and Measures | Loadings |
|---|---|
| Regulatory pressure [13] | |
|    [RP1] Strict compliance with government regulations. | 0.954 |
|    [RP2] Influence of government policy on promoting willingness to implement innovation. | 0.971 |
|    [RP3] The favorable treatment of local governments for implementing innovation. | 0.968 |
|    [RP4] The impact of government funding on innovation. | 0.939 |

**Table 2.** *Cont.*

| Constructs and Measures | Loadings |
|---|---|
| Normative pressure [13] | |
| [NP1] Adoption of environmental products by customers. | 0.890 |
| [NP2] Legitimacy of organizational activities. | 0.889 |
| [NP3] Growing stakeholder awareness of environmental innovation. | 0.921 |
| [NP4] Social media's impact on environmental innovation. | 0.881 |
| Mimetic pressure [13] | |
| [MP1] Responses to stakeholders of the main competitors adopting innovation. | 0.923 |
| [MP2] Main competitors' policy advantages in adopting innovation. | 0.951 |
| [MP3] Increase the ripple effect of main competitors adopting innovation. | 0.956 |
| [MP4] Increase the competitiveness of main competitors that adopt innovation. | 0.930 |
| Green product/service innovation [37] | |
| [GSI1] Choose the least polluting method for product/service development. | 0.854 |
| [GSI2] Choose the method of using the least energy and resources to develop products/services. | 0.865 |
| [GSI3] Configure product/service development with minimal material. | 0.857 |
| [GSI4] Consider recycling and reuse for product/service development. | 0.889 |
| Green process innovation [37] | |
| [GPI1] Recycling waste and emissions during the business process. | 0.850 |
| [GPI2] Effectively reduce emissions of hazardous materials or waste during business processes. | 0.840 |
| [GPI3] Effectively reduce consumption of natural resources during business processes. | 0.846 |
| [GPI4] Effectively reduce the use of raw materials during business processes. | 0.861 |
| Entrepreneurial orientation [105] | |
| [EO1] An aggressive investment toward uncertainty. | 0.907 |
| [EO2] An aggressive attitude toward uncertainty. | 0.907 |
| [EO3] Importance of R&D and technological innovation activities. | 0.868 |
| [EO4] Leading the way in introducing green products, services, or technologies. | 0.915 |
| Economic performance [106] | |
| [ECP1] Profit growth is superior to that of the main industry competitors. | 0.924 |
| [ECP2] Growth in return on investment is superior to that of industry leaders. | 0.940 |
| [ECP3] Growth in return on sales is superior to that of main industry competitors. | 0.929 |
| [ECP4] Market share growth is superior to that of main industry competitors. | 0.936 |
| Social performance [106] | |
| [SOP1] Reduces social inequality (polarization and regional income disparity) compared to its main industry competitors. | 0.961 |
| [SOP2] Contributes to the spread of social values (e.g., labor rights, revitalization of local communities) compared to its main industry competitors. | 0.958 |
| [SOP3] Enhanced worker or community health and safety compared to its main industry competitors. | 0.958 |
| [SOP4] Protected the claims and rights of aboriginal peoples or the local community compared to its main industry competitors. | 0.951 |
| Environmental performance [106] | |
| [ENP1] Reduces its energy consumption compared to its main industry competitors. | 0.959 |
| [ENP2] Reduces waste (e.g., air, water, and/or solid) emissions compared to its main industry competitors. | 0.948 |
| [ENP3] Reduced the environmental impacts of its products or services compared to its main industry competitors. | 0.957 |
| [ENP4] Undertook voluntary actions (e.g., actions that are not required by regulations) for environmental restorations compared to its main industry competitors. | 0.932 |

*4.3. Common Method Bias Assessment*

This study prevents common method bias (CMB) problems through five methods. First, it is emphasized that there are no predetermined correct or incorrect answers to notify the participants of this study that anonymity and confidentiality are sufficiently guaranteed and to encourage honest answers [108]. Second, there is a time difference in creating a

separate questionnaire for independent and dependent variables by asking respondents for a questionnaire on independent variables first and distributing a questionnaire on dependent variables second, after collecting the questionnaire. Third, through Harman's single factor test, the explanatory power of a single factor according to principal component analysis is 34.0%, confirming that it does not exceed the threshold of 50% [109]. Fourth, as suggested by Kock [110], it is reviewed that the value of the variance inflation factor (VIF) between components does not exceed 3.3 (1.000 < all VIF < 2.928). Finally, based on the suggestions by Lindell and Whitney [111], a partial correlation is conducted using cost leadership strategy as a marker variable, and it is found that it is not statistically significant. These findings suggest that CMB does not present a serious concern in our study.

### 4.4. Analysis Method

In this study, the hypothesis is verified by applying the partial least squares structural equation model (PLS-SEM). The PLS-SEM is considered suitable for complex path models and has the advantage of being relatively free from rigorous and unrealistic assumptions (e.g., multivariate normality) and sample size [112]. Our study discusses corporate green innovation and sustainable performance under institutional pressures and considers entrepreneurial orientation as a factor that can influence these mechanisms. Therefore, the analysis should be conducted from an integrated perspective that connects the internal innovation of the firm with external influencing factors. Although PLS-SEM is controversial because it is based on some unrealistic assumptions, we determine that utilizing PLS-SEM is more effective in reliably estimating parameters and verifying integrated causality.

### 4.5. Validity and Hypotheses Tests

Table 3 is the result of an analysis of its validity. First, Cronbach's alpha coefficient for all constituent concepts was between 0.917 and 0.970. Second, the factor weights and factor loading of all variables were significant, and the AVE values of all configurations were found to be between 0.736 and 0.917, confirming that convergence validity was satisfied. In addition, this study achieved discriminant validity, as all AVE estimates were larger than the square of the correlation coefficients between all constructs, and the confidence interval does not show a value of 1 in any variable [113].

**Table 3.** Reliability and validity verification results.

| Construct | Age | Size | Type | 01 | 02 | 03 | 04 | 05 | 06 | 07 | 08 |
|---|---|---|---|---|---|---|---|---|---|---|---|
| Firm age | **1.000** | | | | | | | | | | |
| Firm size | 0.236 | **1.000** | | | | | | | | | |
| Industry type | 0.047 | −0.025 | **1.000** | | | | | | | | |
| 01. RP | 0.001 | 0.017 | 0.003 | **0.958** | | | | | | | |
| 02. NP | 0.031 | 0.068 | −0.030 | 0.735 | **0.895** | | | | | | |
| 03. MP | 0.054 | 0.076 | 0.006 | 0.698 | 0.739 | **0.940** | | | | | |
| 04. GI * | 0.045 | 0.045 | 0.011 | 0.691 | 0.716 | 0.739 | **0.858** | | | | |
| 05. EO | 0.011 | 0.065 | −0.019 | 0.571 | 0.626 | 0.571 | 0.657 | **0.899** | | | |
| 06. ECP | −0.001 | 0.036 | −0.028 | 0.704 | 0.700 | 0.297 | 0.696 | 0.692 | **0.932** | | |
| 07. SOP | 0.007 | 0.027 | −0.037 | 0.674 | 0.707 | 0.702 | 0.715 | 0.692 | 0.808 | **0.957** | |
| 08. ENP | −0.014 | 0.022 | −0.018 | 0.678 | 0.686 | 0.679 | 0.669 | 0.662 | 0.765 | 0.810 | **0.949** |
| Cronbach's alpha | | | | 0.970 | 0.917 | 0.956 | 0.949 | 0.921 | 0.950 | 0.970 | 0.963 |
| rho_a | | | | 0.970 | 0.918 | 0.958 | 0.949 | 0.921 | 0.951 | 0.970 | 0.963 |
| AVE | | | | 0.917 | 0.802 | 0.883 | 0.736 | 0.809 | 0.869 | 0.916 | 0.900 |
| $R^2$ | | | | | | | 0.666 | 0.431 | 0.540 | 0.584 | 0.508 |
| $Q^2$ | | | | | | | 0.628 | 0.425 | 0.555 | 0.569 | 0.525 |
| HTMT < 0.85 | | | | Yes | Yes | Yes | Yes | Yes | Yes | Yes | Yes |

Note: * reflective-reflective second order, the square roots of the AVE values are shown on the diagonals, and printed with bold, non-diagonal elements are the latent variable correlations. All correlations are statistically significant at $p < 0.01$, rho_a = Dijkstra and Henseler's composite reliability, HTMT = heterotrait–monotrait ratio of correlations [114]. RP = regulatory pressure, NP = normative pressure, MP = mimetic pressure, GI = green innovation, EO = entrepreneurial orientation, ECP = economic performance, SOP = social performance, ENP = environmental performance.

Table 4 and Figure 2 present the results of path analysis to verify the hypothesis. First, an analysis of the path that institutional pressures have on green innovation shows that regulatory pressure (β = 0.184, *p* < 0.001), normative pressure (β = 0.175, *p* < 0.001), and mimetic pressure (β = 0.341, *p* < 0.001) all correlate with green innovation with statistical significance. Thus, Hypothesis 1 was supported.

**Table 4.** Significance testing of path effects with bootstrap.

| Path | β | S.E. | *t*-Statistic | *p*-Value | BCCI |
|---|---|---|---|---|---|
| RP→GI | 0.184 | 0.042 | 4.416 | *** | |
| NP→GI | 0.175 | 0.049 | 3.561 | *** | |
| MP→GI | 0.341 | 0.045 | 7.651 | *** | |
| RP→EO | 0.181 | 0.050 | 3.647 | *** | |
| NP→EO | 0.362 | 0.065 | 5.537 | *** | |
| MP→EO | 0.177 | 0.058 | 3.053 | ** | |
| EO→GI | 0.249 | 0.045 | 5.548 | *** | |
| RP→EO→GI | 0.045 | 0.015 | 2.909 | ** | 0.019, 0.084 |
| NP→EO→GI | 0.090 | 0.021 | 4.268 | *** | 0.051, 0.139 |
| MP→EO→GI | 0.044 | 0.017 | 2.611 | ** | 0.016, 0.082 |
| GI→ECP | 0.372 | 0.045 | 8.234 | *** | |
| GI→SOP | 0.399 | 0.050 | 7.993 | *** | |
| GI→ENP | 0.314 | 0.047 | 6.685 | *** | |
| EO→ECP | 0.320 | 0.050 | 6.383 | *** | |
| EO→SOP | 0.266 | 0.046 | 5.768 | *** | |
| EO→ENP | 0.245 | 0.049 | 5.034 | *** | |
| EO x GI→ENP | −0.104 | 0.022 | 4.769 | *** | |
| EO x GI→SOP | −0.126 | 0.024 | 5.169 | *** | |
| EO x GI→ENP | −0.152 | 0.021 | 7.163 | *** | |

Note: All indirect effects are partially mediated, 5000 iterations for bootstrapping, confidence level is 95%, S.E. = standard error, RP = regulatory pressure, NP = normative pressure, MP = mimetic pressure, GI = green innovation, EO = entrepreneurial orientation, ECP = economic performance, SOP = social performance, ENP = environmental performance, ** *p* < 0.01, *** *p* < 0.001.

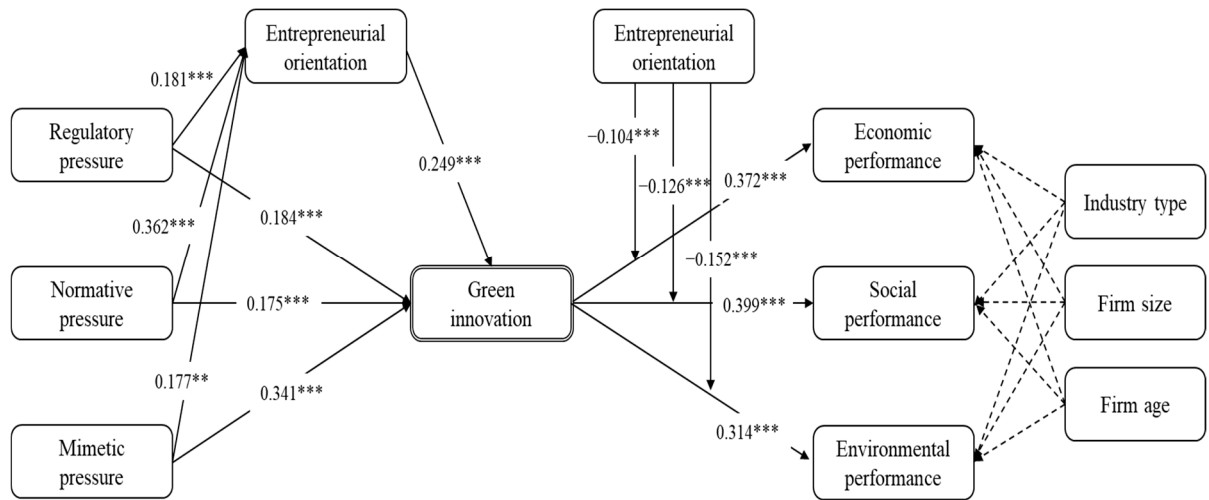

**Figure 2.** Estimated results of a structural equation analysis. Note: Non-significant paths are shown by a dotted line; ** *p* < 0.01, *** *p* < 0.001.

Second, we validate the mediating role of entrepreneurial orientation in the direct impact of institutional pressures on green innovation. The results of 5000 replicates using boot-strapping are as follows [115]. The direct effects between regulatory pressure, normative pressure, mimetic pressure, and entrepreneurial orientation were 0.181 (*p* < 0.001), 0.362 (*p* < 0.001), and 0.177 (*p* < 0.01), respectively. The direct effect between entrepreneurial orientation and green innovation was 0.249 (*p* < 0.001). In addition, the indirect effects

of regulatory, normative, and mimetic pressures on the green innovation path through entrepreneurial orientation were 0.045 ($p < 0.01$), 0.090 ($p < 0.001$), and 0.044 ($p < 0.01$), respectively. As a result of the confidence interval verification, it was found that the bias-correct confidence interval did not contain 0, and the indirect effect was found to be significant at the 5% level. As a result of verifying partial and complete mediation through Baron and Kenny's [116] three-step test, there was a significant partial mediation effect ($p < 0.05$). Thus, Hypothesis 2 was also supported.

Third, there was a positive causal relationship between green innovation and sustainable performance, and Hypothesis 3 was supported. Specifically, green innovation has a positive effect on economic performance ($\beta = 0.372$, $p < 0.001$), social performance ($\beta = 0.399$, $p < 0.001$), and environmental performance ($\beta = 0.314$, $p < 0.001$).

Fourth, the moderating effect of entrepreneurial orientation between green innovation and sustainable performance was negatively significant in all paths. The moderating effect of entrepreneurial orientation was $-0.104$ between green innovation and economic performance, $-0.126$ between green innovation and social performance, and $-0.152$ between green innovation and environmental performance, all of which were significant at the 1% level.

## 5. Discussion and Implications

### 5.1. Discussion

This is a discussion of some main findings found in our study. First, this study predicted a positive relationship between institutional pressures and corporate green innovation, and the empirical results were consistent with discussions of previous studies supporting this hypothesis [4,13,50]. Specifically, regulatory pressure, normative pressure, and mimetic pressure all have a positive and significant effect on green innovation in firms, a result that supports the Porter hypothesis [26]. These findings are consistent with the view of previous studies that institutional pressures will drive green innovation to meet regulatory and normative requirements and enhance competitive advantage rather than hinder corporate innovation. In addition, as the severity of environmental pollution grows, it supports the argument that increasing institutional pressures will drive businesses to implement sustainable supply chain management and promote green production [117].

Second, the positive mediating effect of entrepreneurial orientation between institutional pressure and green innovation supports research from an entrepreneurship perspective that entrepreneurial orientation is a pivotal factor in supporting green innovation and practice in the face of strict environmental regulations. Our empirical evidence shows that institutional pressure not only has a direct positive effect on green innovation through entrepreneurial orientation but also has an indirect positive effect. This shows that institutional pressures are an external driver of entrepreneurial orientation. In addition, it is consistent with the discussion that entrepreneurial orientation is required to achieve green innovation by encouraging firms to challenge innovative environmental practices under institutional pressures. For instance, Alshebami [118] stated that green entrepreneurial orientation positively influences green innovation, which positively mediates the relationship between green entrepreneurial orientation and economic performance. Accordingly, entrepreneurial orientation can contribute to promoting innovation in a firm's products, services, and operations processes in terms of pollution control and energy efficiency by paying attention to environmental issues and being environmentally friendly.

Third, this study hypothesized the moderating role of entrepreneurial orientation between green innovation and sustainable performance. Our empirical finding confirmed that entrepreneurial orientation had a negative moderating effect on green innovation and sustainable performance. One possible reason for this result is that in the absence of sufficient innovation, too-high risk-taking is likely to prevent the firm from leveraging internal resources, which can be a hindrance to corporate performance [99]. March [119] explains that even with a high entrepreneurial orientation, the innovation achieved can be discarded, and even if a firm aggressively pursues innovation opportunities, it is likely

to end exploratory, making it difficult to realize results. Moreover, Tang, Tang, Marino, Zhang, and Li [97] point out that entrepreneurial orientation can rather be a hindrance to performance creation because institutional constraints such as *Quanxi* exist for firms in emerging economies such as China. Therefore, due to these environmental conditions in China, the results of our study may have shown that entrepreneurial orientation rather hinders the relationship between green innovation and sustainable performance.

### 5.2. Theoretical Contributions

First, this study expands prior research on the determinants of green innovation by integrating institutional theory and green innovation literature to identify the link between institutional pressures and green innovation. Second, our study enriches the entrepreneurship literature by examining the mediating effect of entrepreneurial orientation in the relationship between institutional pressures and green innovation and by identifying the diverse roles of entrepreneurial orientation in promoting green innovation in corporate responses to institutional pressures. Third, our results indicate that entrepreneurial orientation acts as a negative moderating effect between green innovation and sustainable performance. Most existing studies highlight the positive effects of entrepreneurial orientation on corporate performance [83,98]. This study integrates sustainable management and entrepreneurship and identifies the heterogeneous effects of entrepreneurial orientation on sustainable performance, represented by economic, social, and environmental performance. By identifying the diverse roles of entrepreneurial orientation, we met the demand for more research on entrepreneurial orientation. In other words, this study goes one step further in terms of the role of entrepreneurial orientation by identifying negative moderating effects, unlike previous literature that expects a positive role of entrepreneurial orientation between corporate innovation and performance. Fourth, our study provides a broad perspective on corporate social responsibility (CSR) context by securing significance in achieving corporate social and environmental performance through green innovation.

### 5.3. Practical Implications

This study has several practical implications. First, our study found that institutional pressures have a significant impact on corporate green innovation, and managers should keep an eye on institutional change and take full advantage of it to move toward successful green innovation. In order to increase sustainable performance in response to institutional pressures, the active participation of managers in green innovation practices (e.g., RE100, Science Based Targets Initiative) should be encouraged. For example, Siemens is recognized as a green leader, as it successfully drives green innovation practices based on energy efficiency, decentralized energy systems, intelligent e-mobility solutions, and green power transactions to achieve full carbon neutrality by 2030 in Germany, where government regulations on the environment are firm. In addition, governments need to play a more pivotal role by strengthening the introduction of environmental protection laws and regulations and by continuing to improve the participation of stakeholders in environmental supervision.

Second, we suggest that when firms take more risks and have entrepreneurial orientation that enables entrepreneurial orientation, they can drive green innovation out of institutional pressures. Developing an entrepreneurial orientation can benefit businesses by helping them proactively respond to external institutional pressures on green technology innovation. In particular, entrepreneurship can create jobs, stabilize society, and have a close connection to solving sustainable problems, so companies should consider operating in-house training programs that can foster entrepreneurial orientation. More importantly, this entrepreneurial orientation should be used as a driving force for innovation by linking it with corporate operations and strategies. GE has emerged as an eco-friendly leader by adopting a new strategy, ecomagination, through its entrepreneurial orientation towards green innovation. Despite these advantages, depending on the business environment a firm is in, entrepreneurial orientation, which tends to take a lot of risks, can hinder sustainable

performance; thus, firms should pay attention to decisions that increase entrepreneurial orientation after green innovation.

Third, our study promotes interdisciplinary discussions on sustainable development by investigating internal and external environmental factors for corporate sustainable management and incorporating different research fields, especially to achieve green innovation. Furthermore, small and medium-sized enterprises (SMEs) may face unique challenges in implementing green innovation due to their limited resources; thus, our research helps to develop effective green innovation pathways for SMEs.

## 6. Conclusions

Based on institutional theory and an entrepreneurship perspective, we explore the role of entrepreneurial orientation among institutional pressures (i.e., regulatory, normative, and mimetic pressures), green innovation, and sustainable performance (i.e., economic, social, and environmental performance). Specifically, by analyzing the impact of institutional pressure on green innovation and identifying the mediating role of entrepreneurial orientation, this study explores the question of why some firms participate in green innovation more than others. It also expands its role and knowledge of entrepreneurial orientation by identifying the moderating role of entrepreneurial orientation between green innovation and sustainable performance. We sampled 483 observations of listed firms in China that had a significant positive effect on institutional pressures and green innovation and found a mediating effect of entrepreneurial orientation. In addition, entrepreneurial orientation confirmed the negative moderating effect of the relationship between green innovation and sustainable performance. This study provides some implications for inconsistent results in both literature and practice, as well as contributions to current knowledge.

Despite these contributions, the study has certain limitations. First, corporate green innovation can be driven by internal as well as external factors, but this study was limited to external institutional pressures as determinants of green innovation. Second, as digital transformation is emerging as a major strategy in corporate response to institutional pressures on green innovation, it is necessary to examine how digitalization factors affect it. In particular, digital transformation has become a notable central trend in corporate approaches to green innovation. Third, since this study may limit its generalizability as it is analyzed through a sample of firms in China, more studies should be conducted in other countries and regions. Finally, this study is based on survey data, which cannot reflect changes in firms. In future studies, longitudinal studies based on time series and/or panel data can be conducted to explore and measure changes in entrepreneurial orientation.

**Author Contributions:** Q.Z. and M.-J.L. contributed to the conceptualization, methodology, investigation, and writing—original draft; Q.Z., X.Z. and M.-J.L. performed research modeling, data collection, data curation, and formal analysis; Q.Z., X.Z. and M.-J.L. participated in the manuscript revision, review, editing, and validation. All authors have read and agreed to the published version of the manuscript.

**Funding:** This research received no external funding.

**Institutional Review Board Statement:** Not applicable.

**Informed Consent Statement:** Not applicable.

**Data Availability Statement:** The data used in results will be provided upon a reasonable request.

**Acknowledgments:** The authors would like to thank the editors and anonymous reviewers for their insightful comments and suggestions.

**Conflicts of Interest:** The authors declare no conflicts of interest.

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
