# Peer review of "Exploring Institutional Pressures, Green Innovation, and Sustainable Performance: Examining the Mediated Moderation Role of Entrepreneurial Orientation"

_sustainability, doi:10.3390/su16052058_

Round 1

Reviewer 1 Report

Comments and Suggestions for Authors

Research designed to understand the multifaceted role of entrepreneurial orientation among institutional pressures, green innovation, and sustainable performance through an institutional theory and entrepreneurship perspective may be of great value to the literature and industry. The study presents arguments coherently, and describes the results in great detail. However, to provide a comprehensive picture regarding the trends in the corporate environment and business innovation it is recommended to refer to the following comments.

1.      The digital transformation is a central trend in the corporate approach to green innovation that deserves attention. Authors are advised to address this trend to emphasize the implications of the study.

2.   Another important trend that deserves attention is corporate social responsibility and environmental performance. It is recommended that the authors refer to CSR in the context of green innovation and corporate performance as part of the discussion and implications.

3.    From a grammatical point of view, it is recommended to maintain the uniformity of the use of initial letters. It is advisable to examine the uniform use of initial letters in subsections 4.2, 4.3, 4.4, 4.5 in the study.

Reviewer 2 Report

Comments and Suggestions for Authors

The authors wrote an excellent paper with research, hypotheses, and a model that clearly demonstrated the assertion of the tested hypotheses. The study aimed to understand the multiple role of entrepreneurial orientation between institutions, namely national pressures, green innovation and sustainable performance, through the skillful use of entrepreneurship theory and perspectives. The literature in the paper is also well cited and truly comprehensive. In addition to the authors analyzing the impact of institutional pressures - regulatory and normative on green innovation, that is, the mediating effect of entrepreneurial orientation, the paper also focused in one part on the mitigating effect of entrepreneurship, which it has on green innovation and sustainable performance. The number of examined companies is also satisfactory. I believe that the paper can be published as it is, without any objections.

Reviewer 3 Report

Comments and Suggestions for Authors

First, this reviewer believes that companies that are said to have contributed the most the climate change crisis should seek to mitigate their responsibility through environmental innovation. Furthermore, as argued in this study, there is also a strong case to be made for sustainable performance for corporate survival through environmental innovation.

This study proposes a somewhat complex model for this purpose, and I think it draws some meaningful conclusions. However, there are a few points we would like to make below.

Major issues

1.     It is preferable to show all the factor loading values for total variables in the table, rather than just one. This work enables cross-loading check between variables.  

2.     The importance of entrepreneurial orientation seems to have been underemphasized in the literature, and there is a need to emphasize the importance of entrepreneurial orientation itself, not just its mediating and moderating roles among variables.

3.     The correlation analysis presented in Table 2 shows that the correlation between ECP and SOP is 0.808, and the correlation between SOP and ENP is 0.810. So, based on this result, can you argue that these are distinct concepts?

4.     Despite the presentation and analysis of very complex research model, the discussion and implications of the paper seem to be rather brief. The findings deserve a full discussion.

Minor issues

1.     Table 2. Reliability and validity verification resultsà Table 3

2.     When interpreting statistical results, keep the following in mind statistical significance levels should be specified by the researcher, not by the program.

For example, b=0.184, p<0.000-à β=0.184, p<0.01.

3.     Pls-sem may be an effective methodology for analyzing complex models as suggested by the researcher, but it is still a controversial methodology because it is based on some unrealistic assumptions.  

Comments on the Quality of English Language

Double-check your grammar and context.

Reviewer 4 Report

Comments and Suggestions for Authors

Thank you for the chance to evaluate this article. Please consider the following points:

1. This article appears compelling as it integrates institutional theory with various

entrepreneurial constructs and environmental factors, all of which are currently significant

topics.

2. I'm curious about the use of the term "pressure." Social support doesn't necessarily imply

pressure in institutional theory. Positive support from society, family, and peers for a certain

action may not be considered pressure. The use of this term might suggest an initial assumption

of negative pressure only.

3. If you are formulating more than one hypothesis (sub-hypotheses), it would be beneficial to

depict this in your proposed model to enhance clarity for the reader.

4. I recommend incorporating insights from the article "Green Innovation, Self-Efficacy

Entrepreneurial Orientation, and Economic Performance: Interactions among Saudi Small

Enterprises (2023)" to strengthen your discussion section.

Please show the VIF values.

5. Ensure consistent use of the term "corporate" throughout the article.

Wishing you success in your work.

Comments on the Quality of English Language

needs proofreading 

Reviewer 5 Report

Comments and Suggestions for Authors

Introduction   Explain the Research Objectives: Although the article outlines the context and importance of the research, it could be useful to define the specific research objectives more explicitly at the beginning of the introduction. This would help orient the reader to what the study seeks to achieve or discover.  

Highlight the Novelty of the Study: While the introduction addresses the importance of the study, emphasizing more clearly what makes the research unique from previous studies could strengthen the argument for its necessity. This includes highlighting any specific method, theory or context that is particularly innovative.  

Theoretical Background Expand References and Comparatives: While the section offers a good overview of the key concepts, it could benefit from including a wider range of references and comparative studies to highlight current debates and divergences in the literature. This would help situate the research within the wider academic context, showing how it aligns with or differs from previous work.  

Contextualize Concepts Further: While the section does a good job of defining key concepts, it could benefit from a deeper contextualization of how these concepts apply specifically to the Chinese context, considering China's institutional, cultural and economic particularities.   

Integration between Concepts: A more holistic approach would help readers understand how these theoretical elements combine to form the study's theoretical framework; it would be interesting to represent this relationship using a figure.  

Hypothesis Development   The clearer connection between hypotheses and existing literature: Although the section references previous studies to justify the hypotheses, there could be a deeper and more explicit integration with existing literature. This includes detailing how the proposed hypotheses align with or diverge from previous findings, providing a more solid basis for each hypothesis.  

Empirical justification for specific hypotheses: While some hypotheses are justified because of previous studies, others could benefit from stronger empirical justification. This could include discussing specific empirical evidence supporting the proposed relationships.  

Discussion of the cultural and institutional context: Considering that the study is contextualized in China, a more in-depth discussion of how the country's specific cultural and institutional context may influence the relationships.  

Clarity and precision in formulating the hypotheses: While generally clear, some could be formulated more precisely to avoid ambiguity. This includes being specific about the nature of the expected relationships (e.g., positive, negative, moderating) and the specific constructs involved.   To improve the representation of Figure 1, it only represents some of the hypotheses.  

Research Context and Sample Collection   A more detailed description of sample selection and data collection process: Although the section mentions a questionnaire survey with companies listed in China, a more detailed description of the sample selection process and response rate could improve the transparency and replicability of the study.  

In-depth explanation of data processing and cleaning: Information on how the data was processed and cleaned before analysis could clarify potential concerns about bias or errors in the data.  

Greater clarity in justifying the choice of analysis method (PLS-SEM): Although the authors mention why they chose PLS-SEM, a more in-depth discussion of the advantages and limitations of this method over other approaches could strengthen the methodological basis.  

Visual presentation of the results: Although the results are presented in tables, including graphs or visual models could help interpret and visualize the relationships between the variables.  

Discussion   Consideration of Contextual Variables: Discuss how contextual variables, such as cultural, economic and regulatory differences, may influence the generalizability of the results to other contexts or regions.

  More profound Comparative Analysis: Although the section already discusses existing literature results, a deeper comparative analysis with similar studies could enrich the discussion by highlighting where the results align with or diverge from previous research. This would help identify specific nuances in the results and provide insights into possible reasons for any discrepancies.  

Theoretical Contributions   Explain Implications for Institutional Theory: It could be useful to detail how the study's specific findings propose extensions or refinements to institutional theory, especially in the context of the research.  

Deeper Connection to Entrepreneurship Literature: Although the section mentions the contribution to entrepreneurship literature, a more in-depth discussion of how the results align with or diverge from existing theories of entrepreneurial orientation could enrich the analysis.  

Discussion of Interdisciplinarity: A more detailed exploration of how the study integrates and contributes to various research areas, such as innovation management, sustainability and entrepreneurship, would highlight the relevance and applicability of the findings in interdisciplinary contexts.  

Practical Implications   Detailing Implications for Managers: Although the section already provides some guidance for managers, further detailing how they can apply the study's findings to concrete strategies would help make the implications more tangible. For example, explaining specific steps managers can take to increase green innovation in response to institutional pressures could be useful.  

Including Case Studies or Practical Examples: Incorporating case studies or examples of companies that have successfully implemented green innovations in response to institutional pressures or have cultivated an effective entrepreneurial orientation could illustrate how the implications can be applied in practice.  

Advice for small and medium-sized enterprises (SMEs): Given that SMEs can face unique challenges in implementing green innovations due to limited resources, providing specific guidance for these companies could make the section more inclusive and practical.  

Implications for Management Education: Suggesting how management and entrepreneurship education programs can incorporate the study's findings to prepare future business leaders to face environmental challenges would also be a valuable addition.  

Conclusion   Relation of the Results to the Study Objectives: A more explicit discussion in the introduction of how the results meet the specific objectives could clarify the fulfillment of the study's goals.  

Limitations and Future Research Directions: Although the section addresses some limitations and suggests future research directions, further detailing these aspects, including potential methodologies, contexts, or variables to be explored, could provide a clearer roadmap for future work.  

Clarity and Conciseness: Ensuring the conclusion is clear, concise and free of unnecessary technical jargon would help make it accessible to a wider audience, including those outside the specific academic field.

Round 2

Reviewer 3 Report

Comments and Suggestions for Authors

We believe that the fixes requested by this reviewer have been fulfilled. Thank you for your efforts. 

Reviewer 4 Report

Comments and Suggestions for Authors

satisfied

Reviewer 5 Report

Comments and Suggestions for Authors

Good job!